

# An investigation into the bilateral functional differences of the lower limb muscles in standing and walking

Shengyun Liang[1], Jiali Xu[2], Lei wang[1] and Guoru Zhao[1]

[1] Shenzhen Institutes of Advanced Technology, Chinese Academy of Sciences, Shenzhen, China
[2] Mechanical and Control Systems Engineering Department, INSA of Rennes, Rennes, France

## ABSTRACT

To date, most studies use surface electromyographic (sEMG) signals as the control source on active rehabilitation robots, and unilateral data are collected based on the gait symmetry hypothesis, which has caused much controversy. The purpose of this study is to quantitatively evaluate the sEMG activity asymmetry of bilateral muscles in lower extremities during functional tasks. Nine participants were instructed to perform static and dynamic steady state tests. sEMG signals from the tibialis anterior, soleus, medial gastrocnemius and lateral gastrocnemius muscles of bilateral lower extremities were recorded in the experiments. Muscle activities are quantified in terms of sEMG amplitude. We investigated whether characteristics of left limb and the one of the right limb have the same statistical characteristics during functional tasks using The Wilcoxon rank-sum test, and studied dynamic signal irregularity degree for sEMG activities via sample entropy. The total of muscle activities showed significant differences between left limb and right limb during the static steady state ($p = 0.000$). For dynamic steady states, there were significant differences for most muscle activities between left limb and right limb at different speeds ($p = 0.000$). Nevertheless, there was no difference between the lateral gastrocnemius for bilateral limb at 2.0 kilometers per hour ($p = 0.060$). For medial gastrocnemius, differences were not found between left limb and right limb at 1.0 and 3.0 kilometers per hours ($p = 0.390$ and $p = 0.085$, respectively). Similarly, there was no difference for soleus at 3.0 kilometers per hour ($p = 0.115$). The importance of the differences in muscle activities between left limb and right limb were found. These results can potentially be used for evaluating lower limb extremity function of special populations (elderly people or stroke patients) in an objective and simple method.

## INTRODUCTION

Studies show that the kinesitherapy has an obvious rehabilitation effect for dyskinesia patients. Furthermore, the training effect is more obvious in patients with a seriously affected gait function whose treatment started earlier. This kind of therapy is based on the plasticity ability of the neural system, i.e., the ability of the brain central nervous system to reorganize itself. Therefore, rehabilitation training can help restore functions of a damaged central nervous system (*Carr & Shepherd, 1987*). Rehabilitation robots can help alleviate the

Corresponding author
Guoru Zhao, gr.zhao@siat.ac.cn

contradiction between the huge patient population and an insufficient number of physical therapists. As a frontier of rehabilitation engineering, exoskeleton rehabilitation robotics has become one of the hottest research areas. With the development of rehabilitation equipment and bio-sensing technologies, exoskeleton rehabilitation robots have developed from a passive type to an active type. Specifically, surface electromyographic (sEMG) has become a popular choice to control prosthesis for its advantages in being non-invasive, relatively portable and simple (*Kawamoto et al., 2003*).

*He, Kuiken & Lipschutz (2009)* used foot pressure and 11 groups of EMG signals to identify user locomotion modes, such as normal walking, spanning obstacle, going up or down stairs and turning round. *Au, Bonato & Herr (2005)* proposed a method to control the position of active ankle prosthesis based on sEMG. *Cheron et al. (2003)* used a dynamic recurrent neural network to analyze the relationship between EMG signals and the joint angle of lower extremities, then predicted the knee angle based on a pattern recognition model of EMG.

The researches based on the gait symmetry hypothesis are still debated today. Some researchers have proved the existence of the symmetry of the bilateral lower extremities through comparison of their kinetic and kinematic data. *Hannah, Morrison & Chapman (1984)* proved that both the movements of bilateral hips in three planes and the movements of bilateral knees in the sagittal plane were symmetrical by using frequency domain analysis. *Hamill, Bates & Knutzen (1984)* also found that there were no significant differences between the ground reaction forces of bilateral lower extremities during walking and running. However, some researchers agree that bilateral gait is asymmetrical due to the different functions of bilateral lower extremities (*Seeley, Umberger & Shapiro, 2008*; *Sadeghi et al., 2000*). In addition, some researchers pointed out that the performances of bilateral lower extremities were asymmetrical in an energy efficiency aspect (*Arsenault, Winter & Marteniuk, 1986*; *Gundersen et al., 1989*; *Ounpuu & Winter, 1989*).

In this article, an objective and simple method for evaluating the asymmetry of lower limb extremity function is proposed. Firstly, the signals of corresponding muscle activities are extracted from static and dynamic steady state tests. Then, the goal of our research is to study whether different limbs could perform difference in the characteristics of selected muscle activities. This is, to investigate whether characteristics of left limb and the corresponding ones of right limbs have similar statistical characteristics during functional tasks. In addition, we will study the dynamic signal irregularity degree for selected muscle activities at different walk speed.

## EXPERIMENTAL DESIGN

Nine healthy subjects volunteered to participate in the study (mean age: 26.4 years, maximum age: 33 years, minimum age: 24 years). All participants had no lower extremity joint injuries within a year, and did not have intense muscular activities at least 24 h before the experiment. All participants were right-footed (*Wikipedia, 2001*), i.e., they all preferred to use right leg when playing football. The experimental procedure was approved by the Institutional Review Borad of Shenzhen Institutes of Advanced Technology (Reference No.SIAT-IRB-160315-H0103). Each subject signed informed consents prior to testing.

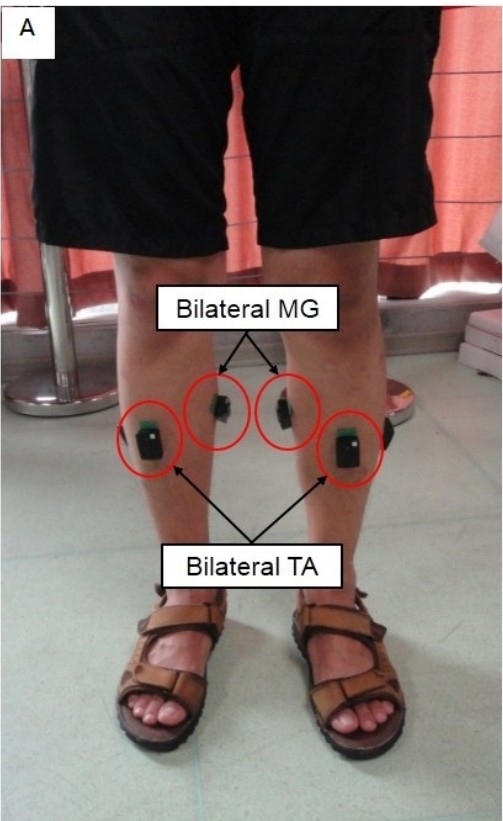
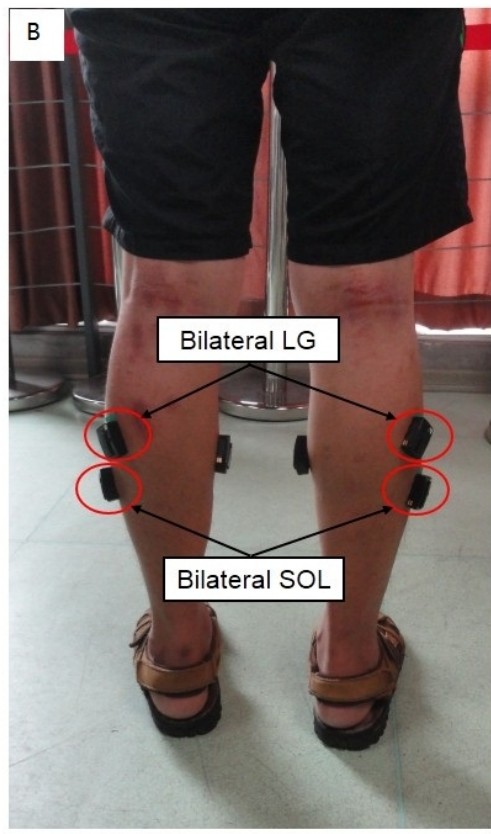

**Figure 1** **Schematic diagram of standing scenario.** (A) The locations of sensors stuck on bilateral TA and MG; (B) The locations of sensors struck on bilateral SOL and LG.

## Data collection and processing

The sEMG signals were collected using 16-channel Trigno wireless EMG system (Delsys Inc., Boston, MA, USA). For our experiments, we adjusted the system to work with the following set of parameters: sampling rate 2000 Hz, band-pass frequency 10–500 Hz, gain 300. The system is provided with a 50 Hz notch filter to eliminate the power frequency disturbance. The sEMG signals were collected from bilateral lower extremity corresponding muscles. Generally, there are nine pairs of muscles form the walking function and each of them contributes in a different way to it. In this research, four pairs of muscles were examined: bilateral tibialis anterior (TA), bilateral medial gastrocnemius (MG), bilateral lateral gastrocnemius (LG) and bilateral soleus (SOL). In a unilateral lower extremity, nine muscles form the walking function and each of them contributes in a different way to it (*Zajac, Neptune & Kautz, 2003*; *Zajac, Neptune & Kautz, 2002*). We employed the method described in *JoNhagen et al. (1996)* to choose the sensor location on the limbs and collect the MVC (Maximum Voluntary Contraction) EMG of each muscle. We used the EMG works 4.0 software from DELSYS Company to do RMS conversion. After that, all the data were normalized by MVC EMG.

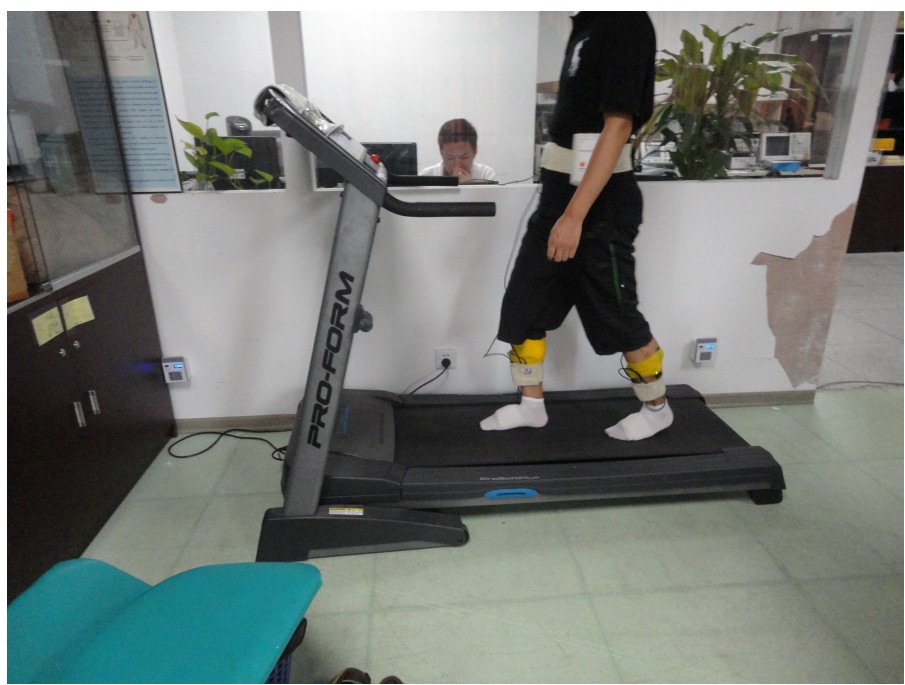

**Figure 2** Schematic diagram of walking scenario.

## Experimental setup
### Static steady state test
The subject was asked to stand straight on a horizontal ground for 1 min with hands naturally down on both sides of the body and to keep shoulder width distance between his/her feet. The recording of the sEMG signals was then performed during standing (Fig. 1).

### Dynamic steady state test
The subject was asked to walk on a treadmill (Fig. 2). After the test started, the speed was increased gradually from 1 km/h to 4.5 km/h with a step of 0.5 km/h, and there was no pause during the test. For each speed, after achieving a state of uniform motion steady speed, recording of sEMG signals was performed. The duration of recording for each speed was limited to 15 s in order to minimize the effects of fatigue.

Periodicity is a major feature of gait. The gait cycle is defined as the time interval between two successive occurrences of one of the repetitive events of walking. It generally uses the time when one of the feet contacts the ground as the beginning of the cycle. If it is decided to start with initial contact of the right foot, then the cycle will continue until the right foot contacts the ground again. However, this definition is not convenient for this research which mainly considers the muscular activity. Therefore, in order to facilitate the analysis, we redefined the gait cycle like this: during normal walking, the onset of the gait cycle is considered to be the contraction of the right TA; the cycle will continue until the right TA contracts again. The corresponding relationship between the gait and the division of the gait cycle is shown in Fig. 3A.

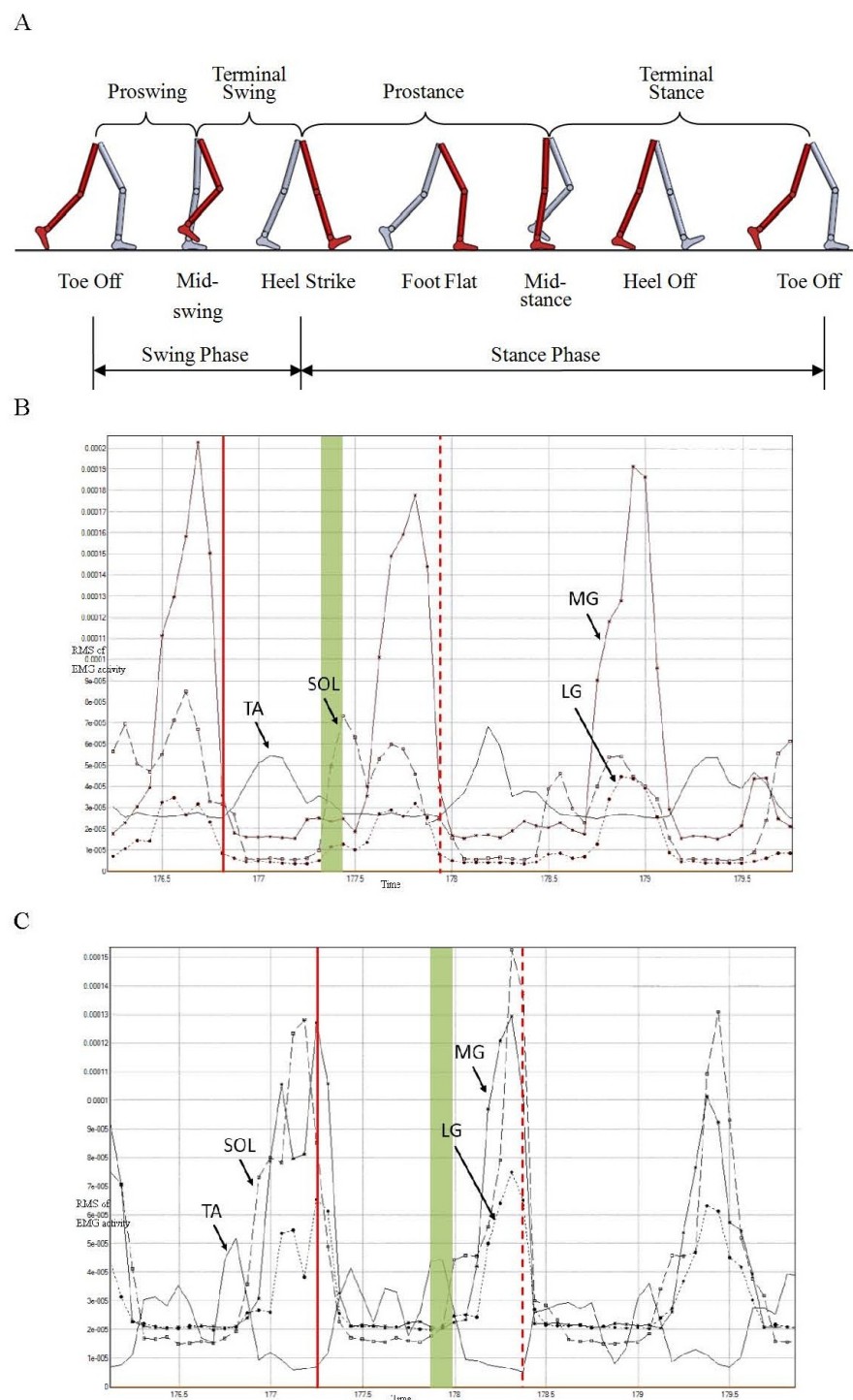

**Figure 3** **Illustration of division of gait cycle (A), RMS of EMG activity of four muscles from right (B) and left (C) lower extremity of one subject while maintaining a speed of 3 km/h.** Verticle solid and dash lines represent starting and termination time of gait cycle, respectively.

Figures 3B and 3C show the Root Mean Square (RMS) curves of EMG activity of four muscles from right and left limb separately at speed of 3 km/h. The time interval between two vertical lines is the gait cycle defined by the new method (Figs. 3B and 3C). The vertical solid lines represent the beginning of the gait cycle, and the vertical dash lines represent the termination of the gait cycle. It was shown that bilateral TA acted during the swing phase and the other three sorts of muscles acted during the stance phase. Moreover, the above-mentioned four muscles of each of the lower limbs acted simultaneously at the moment of transition between swing phase and stance phase; these are indicated by the green filled areas in Figs. 3B and 3C. The overlap between activity of the bilateral TA and SOL is obvious, in particular.

Figure 4 show the ensemble average curves of eight muscles of all subjects in a gait cycle at series of speed. It's indicated that TA is used to control the posture of the foot and it is mainly activated during the swing phase (Figs. 4A and 4B) during normal walking. The actions of SOL to prevent excessive forward movement of the body are the flexion of calf and plantar flexion of foot. Therefore, SOL mainly activates during the stance phase (Figs. 4C and 4D). MG and LG muscles mainly act during the post-stance phase (Figs. 4E–4H). In order to investigate the statistical difference and dynamics signal irregularity degree among sEMG activities of left limb or right limb at series of speed, we will extract one hundred points at equal intervals of each curve on the Fig. 4.

## METHODS

### Sample entropy

Sample Entropy (SampEn) is proposed by Richman and Moorman (*Richman & Moorman, 2000*) and a nonlinear measurement way to analysis time series which can profoundly understand biological system. Lower sample entropy represents more self-similarly of time series. Conversely, higher value indicates more random of time series. Sample entropy is used to measure the time series of EMG activity. The computational procedures of sample entropy are as follow:

Given a standardized (with zero mean and unit variance) time series $\{x_i, i = 1, \ldots, N\}$, $N$ is the total number of data points.

Step1    Construct continuous subsequences of length m: $X_m(1), X_m(2), \ldots, X_m(N - m + 1)$. Where, $X_m(i) = [x_i, x_{i+1}, \ldots, x_{i+m-1}]$, m is called as embedding dimension.

Step2    Define the distance between $X_m(i)$ and $X_m(j)$, given by:

$$d(X_m(i), X_m(j)) = max\{|x_{i+k} - x_{j+k}|; 0 \leq k \leq m - 1\}, 1 \leq i, j \leq N - m + 1, i \neq j.$$

Step3    Calculate the number that any vector $X_m(j)$ which are similar to $X_m(i)$ within r as:

$$Num_m(r) = \Sigma_{i=1, i \neq j}^{N-m+1} \Sigma_{j=1}^{N-m+1} I\{d(X_m(i), X_m(j)) < r\}.$$

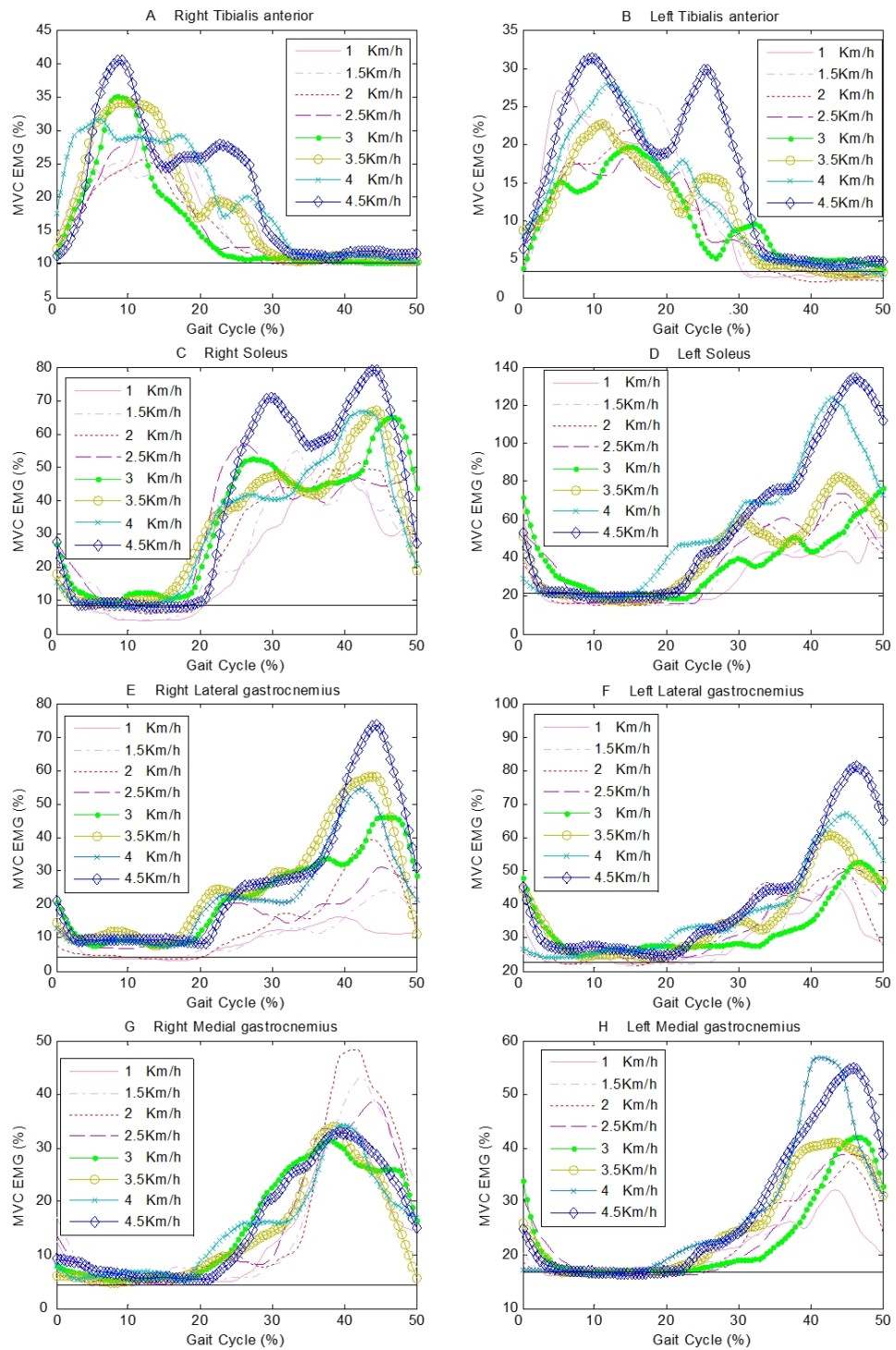

**Figure 4** **The ensemble average curves of 8 muscles of all subjects in a gait cycle at a series of speeds.** The horizontal solid line represents the muscular activities during static standing. (A) right TA, (B) left TA, (C) right SOL, (D) left SOL, (E) right LG, (F) left LG, (G) right MG, (H) left MG.

Step4        Calculate match probability: $\phi_m(r) = \frac{Num_m(r)}{\binom{N-m+1}{2}}$.

Step5        Set $m = m+1$ and repeat steps 1–4.

Step6        Calculate sample entropy as:

$$SampEn = -log\frac{\phi_m(r)}{\phi_{m+1}(r)}.$$

It is important to point out that, in the process of calculating sample entropy, the parameter m and r should be determined. Usually, the optimal value of $m$ is 1–2, $r$ value range from 0.1SD to 0.25SD (SD is standard deviation of time series) (*Aboy et al., 2007*). In our experiment, we selected recommended values $m = 2$ and $r = 0.2$ for experimental data.

## Statistical analysis

The Wilcoxon rank-sum test was developed by Wilcoxon in 1945 and is a nonparametric test without assuming that distribution of each group is normally distributed. This test could assess for significant differences on continuous or discrete variable and be used to analyze the differences among variables of EMG activity between the left and right legs. We used SPSS statistical software for these analyses. The significant level used for identifying difference was $p$-value $< 0.05$.

## RESULTS

### Comparison for sEMG activities between left limb and right limb during the static steady state

When the subject is standing straight, the ankle joint involves only two planes (sagittal and coronal planes). All the selected muscles in the experiments contribute in maintaining the body's static stability.

Mean and standard deviations, pairwise comparisons for selected muscle features for left limb and right limb are shown in Fig. 5. Significant results are indicated with an asterisk (*). There were significant differences in pairwise comparisons of MVG EMG of TA, SOL, LG, MG between left limb and right limb ($p = 0.000$).

As Fig. 5 shows, the mean of MVC EMG of tibialis anterior in left limb was higher than corresponding value in right limb. With regard to soleus, medial and lateral gastrocnemius, the values in right limb was higher than those in left limb. Quantitatively, the MVG EMG of right TA was 2.9 times the size of the one of left TA, and the MVG EMG of left SOL, MG, and LG were 2.5, 5.1 and 1.6 times the size of corresponding muscles at the right side, respectively.

### Comparisons and dynamics signal irregularity degree for sEMG activities during the dynamic steady state

The result of the Wilcoxon rank-sum test for selected muscle acitvities was given in Table 1. For most muscle activities, there were significant differences between left limb and right limb at different speeds. Nevertheless, there was no different bwteen LG for left limb and

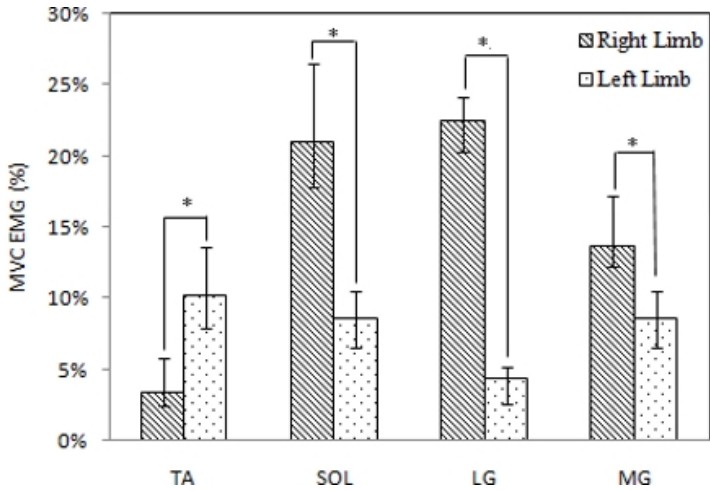

**Figure 5** **Comparisons of means and standard deviations of MVC EMG of all subjects during the static steady state.** * Significant difference of the muscle activities.

**Table 1** **Wilcoxon rank-sum test for muscle activities of bilateral limbs during the dynamic steady state in different speed.**

| | | Speed/(km/h) | | | | | | | |
|---|---|---|---|---|---|---|---|---|---|
| | | 1.0 | 1.5 | 2.0 | 2.5 | 3.0 | 3.5 | 4.0 | 4.5 |
| LG | p-value | .000 | .000 | .060 | .000 | .000 | .000 | .000 | .000 |
| MG | p-value | .390 | .003 | .000 | .011 | .085 | .003 | .005 | .002 |
| SOL | p-value | .000 | .000 | .000 | .000 | .115 | .000 | .000 | .000 |
| TA | p-value | .000 | .000 | .000 | .000 | .000 | .000 | .000 | .000 |

LG for right limb at 2.0 kilometers per hour ($p = 0.060$). For MG valiable, differences were not found between the left limb and right limb at 1.0 and 3.0 kilometers per hour ($p = 0.390$ and $p = 0.085$, respectively). In addition, there was no difference for SOL between the left limb and right limb at 3.0 kilometers per hour ($p = 0.115$).

Mean and standard deviations as well as pairwise comparisons for selected muscle features during the dynamic steady state were shown in Fig. 6. Significant results were indicated with an asterisk (*) via the Wilcoxon rank-sum test. Figure 6 also showed the value of MVG EMG was in different speed. It was indicated that as the speed increased, the muscular activities also increased. It was obvious that the activity level of left SOL was higher than the one of right SOL all the time, and the maximum difference between them was $\triangle d = 21.37\%$, the smallest difference was $\triangle d = 5.23\%$ (Fig. 6C). Similarly, the activity level of right TA was higher than the one of corresponding left muscle, and the maximum difference between them was $\triangle d = 7.76\%$ (Fig. 6A). The activities of bilateral SOL were closer in low-speed dynamic steady state, but as the speed increases, the difference gradually became larger, up to maximum of 25.29% (Fig. 6B). The activity level of left MG was 1.6 times higher compared to the one of corresponding right muscle during static steady state,

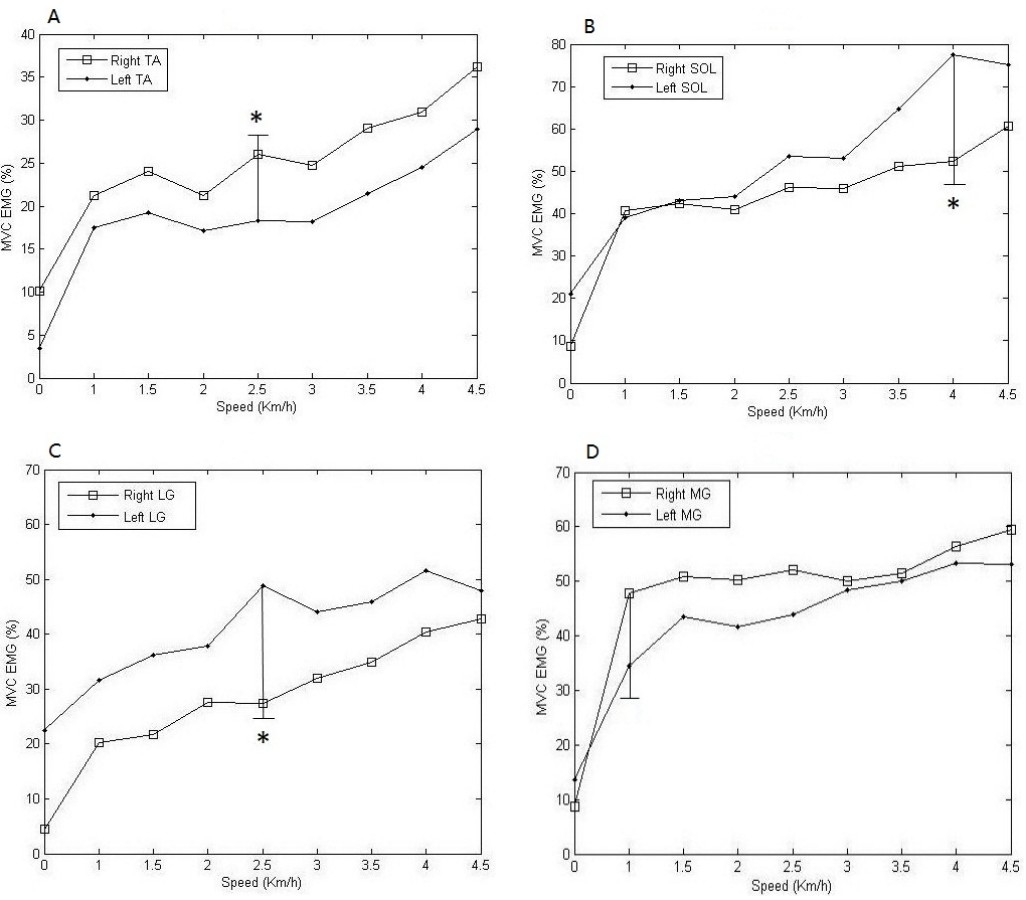

**Figure 6** **Comparisons of means and standard deviations of MVC EMG of all subjects during the dynamic steady state.** The activity levels varied along with the speed change. (A) Bilateral TA; (B) Bilateral SOL; (C) Bilateral LG; (D) Bilateral MG. * Significant difference between muscle activities of the corresponding muscles.

but during dynamic steady state the activity level of right MG was higher than the activity of left MG all the time, and the maximum difference was $\triangle d = 13.27\%$ (Fig. 6D).

In this article, sample entropy is used to measure the dynamics signal irregularity degree. In Table 2, sample entropy values were listed for eight muscles of the left limb and right limb. At different speeds, active degrees of different muscles were significant different between left limb and right limb. It was obvious that the values of the sample entropy of right MG were higher than the corresponding values of left MG at each speed.

## DISCUSSION

### Function of different limbs result in different EMG activities

Some studies have shown the differences existed in kinematic, kinetic and EMG parameters between left and right legs (*Sadeghi et al., 2000*). It is said that bilateral lower limbs have different functions in gait. The right limb is mainly used to provide propulsion while the key role of left limb is to maintain stability of the body and slightly contribute to the propulsion (*Hirokawa, 1989*; *Sadeghi, 2001*). Bilateral lower extremities would exhibit
**Table 2 Sample entropy estimate between variables of the left limb and right limb.**

| | | Speed/(km/h) | | | | | | | |
|------|-----|--------|--------|--------|--------|--------|--------|--------|--------|
| | | 1.0 | 1.5 | 2.0 | 2.5 | 3.0 | 3.5 | 4.0 | 4.5 |
| Left | LG | 0.0324 | 0.0354 | 0.0403 | 0.0395 | 0.0365 | 0.0503 | 0.0537 | 0.0341 |
| | MG | 0.0402 | 0.0479 | 0.0419 | 0.0386 | 0.0429 | 0.0543 | 0.0347 | 0.0409 |
| | SOL | 0.0581 | 0.0728 | 0.0694 | 0.0449 | 0.0689 | 0.0585 | 0.0528 | 0.0589 |
| | TA | 0.0421 | 0.0299 | 0.0202 | 0.0202 | 0.0202 | 0.0202 | 0.0433 | 0.0384 |
| Right | LG | 0.0471 | 0.0492 | 0.0560 | 0.0467 | 0.0309 | 0.0448 | 0.0231 | 0.0202 |
| | MG | 0.0626 | 0.0588 | 0.0563 | 0.0472 | 0.0553 | 0.0628 | 0.0415 | 0.0573 |
| | SOL | 0.0725 | 0.0694 | 0.0684 | 0.0525 | 0.0519 | 0.0493 | 0.0555 | 0.0516 |
| | TA | 0.0337 | 0.0271 | 0.0343 | 0.0364 | 0.0202 | 0.0202 | 0.0202 | 0.0268 |

different patterns of the complex movement, and this will result into difference in each muscle between left limb and right limb.

During a gait cycle, the calves play a role in control and propulsion. The positions of lower limbs should be adjusted in order to insure that there is enough clearance between the foot and ground during the swing phase. This is achieved by collaboration of hip, knee and ankle. For example, the dorsiflexion of ankles caused by activity of TA can adjust the ankle angle for clearance and bear the weight of the body (postural control). During the transition from swing phase to stance phase, the motion speed of the foot was reduced by the eccentric contraction of TA for decelerating the plantar flexion of the foot. In order to prevent the excessive forward toppling of the body, the activity of SOL was significant at this moment. The functions of muscles usually are categorized into motor and stabilizing based on the observed behavior of specific muscles. According this categorization, TA is defined as stabilizing muscle, while the LG has motor function (*Wuebbenhorst & Zschorlich, 2011*). Based on the measurement of energy outputs from knee and ankle during steady walking, *Winter (1983)* found that the propulsion was caused by the plantar flexion of the ankle. *Gottschall & Rodger (2003)* verified this conclusion in their study. Therefore, during the transition, the activities of MG and LG serve for balancing the ground reaction force by generating a certain amount angle of plantar flexion. Besides that, both of them have significant contributions in providing propulsion during the whole stance phase, especially at the end of this phase. Although TA also makes contribution to plantar flexion, but it didn't burst in that moment. This result corresponds with the one of study (*Karlik, Tokhi & Alci, 2003*). The activity of SOL exists throughout the whole stance phase because it can flex the calf and lift the heel. Therefore, complexity of TA, LG, MG and SOL could indicate the functional difference of the lower extremities.

## Difference in different EMG activities during static steady state

Standing straight is one of the most common and important postures in daily life. However, this posture would not be concerned until its key role found by Diener, who investigated the control principle of the body posture control system (*Diener et al., 1984*). For simplicity, human body during standing can be schematically represented as an inverted pendulum.

*Fitzpatrick, Burke & Gandevia (1994)* indicated that the ankle sway directly causes an increase of the postural sway, so the stability of lower extremities plays a key role in maintaining the stability of the whole body. Actually, when standing straight, the body keeps swaying in a micro-range and this is not a completely static state. This means that the location of COP (center of pressure) changes continuously. There are several kinds of systems in the human body that ensure perceiving the postural sway, such as the vestibular, visual and proprioceptive systems. The sensing thresholds of these systems for velocity and amplitude of body sway were determined (*Fitzpatrick & Mccloskey, 1994*; *Simoneau et al., 1996*). The researchers found that the vestibular system, especially the otolith, could determine the acceleration of postural sway both in the sagittal and coronal planes. Generally, when standing straight, the amplitude and velocity of head sway both in sagittal and coronal planes could not reach the thresholds of those systems (*Winter et al., 1998*). With regard to the functions of lower limbs, it shows that the left limb makes a greater contribution for standing. Similarly, we found that there were significant differences in sEMG parameters between left and right limbs.

## Difference in different EMG activities during dynamic steady state

A gait cycle in normal walking generally can be divided into swing and stance phases, where the stance phase usually takes 60% and the swing phase takes 40% (Fig. 3A). The ratio of swing phase to stance phase would become larger with increased walking speed. When achieving the transition speed at which a person changes gait from walking to running, the period of double limb stance would disappear; meanwhile, the ratio could be approximately 60/40 (*Ounpuu, 1994*).

With an increase in walking speed, the period of TA activities increased, while the periods of SOL, LG and MG activities are reduced (Fig. 4). A previous study demonstrated that the amplitude of EMG activities increased as walking speed increased (*Murray et al., 1984*), which are compatible with our study. However, the amplitudes of EMG from all muscles were increasing because demands for maintaining postural stability and providing propulsion were gradually increasing (Fig. 6). Nevertheless, there were varying degrees of differences between the activities of corresponding muscles of bilateral lower extremities at each speed, in accordance with the findings of *Yanagawa et al. (2002)*'s study, which indicated that the muscle activities for elderly people showed significant differences during free, slow and fast walking.

With regard to the EMG activities, if the Wilcoxon rank-sum test shows that some variables have significant statistical characteristics, then those can be independently used for evaluation of lower extremities. Statistical differences were found for most EMG activities. The above result seems similar with the one observed by *Arsenault, Winter & Marteniuk (1986)* that found asymmetries existed in the soleus muscle using EMG data. Considering the differences between EMG activities of the left limb and right limb, the data of the corresponding variables should be collected independently for analysis.

In this study, it has been found that sample entropies of right MG were higher than those of left MG at each speed. It seems that the dynamic signal irregularity degree of MG on the right limb can exhibit more complexity than the one of the left limb. The possible

reason is that a functional asymmetry exist between right limb and left limb. The right limb is responsible for propulsion while the left limb is in charge of supporting the body. Since TA have significant contributions in providing propulsion, its sample entropy on the right limb may provide more discriminative information.

Compared with the dynamic steady state, the static steady state does not need propulsion. Therefore, the steady state is only involved in the control function of the lower extremities in this state. The motor system that consists of the bilateral lower extremities should not only be treated as a simple superposition of two branch chains, but also as a parallel system. Physical stability is achieved by collaboration of both lower limbs.

The muscular activities of TA, SOL and MG from one of the legs were stronger than those of the corresponding muscles from the other leg all the time during the dynamic steady state. The same behavior was observed during static steady state. The interrelation between the responses of the bilateral MG changed during the transition from dynamic to static state. In addition, the result that activity of right MG is stronger than the one of left MG is in agreement with some recent studies (*Ounpuu & Winter, 1989*). This means that the right limb plays a key role for providing propulsion during forward walking.

### Limitation

Our study has several limitations. We redefined the gait cycle in own way while disposing those sEMG signals, which has not been previously investigated. In addition, we only recruited nine healthy individuals who were able to walk independently in normal and relaxed gait, and did not consider special populations. In order to obtain more information for assessing lower limb extremity function, we will increase the sample size at runtime, including elderly people, stroke patients or individuals who are too frail and easily fall down.

## CONCLUSIONS

The purpose of this study was to investigate the quantitative difference between the muscular activities of the bilateral lower extremities and to show the importance of this difference. Nine healthy subjects consented to participant in our study. The experiment included two kinds of situations, i.e., static steady state (standing straight) and dynamic steady state (walking). Muscle function, muscular recruitment pattern and activity ability of corresponding muscles were compared under two states.

Current research work showed that: (1) The corresponding muscle activities between the left limb and right limb showed statistical difference. Therefore, the collection of EMG signal data between left limb and right limb should be separated to acquire the best experimental result. (2) In dynamic steady state test, it's most obvious characteristic was the muscle activities change with the speed of walking. At different speeds, high correlations were found among unilateral or bilateral muscle activities.

In the future, the results could be used to set the control threshold of active robots for rehabilitation of bilateral lower extremities, and could allow precise EMG-based rehabilitation assessment for patients.

### Funding

This study has been financed partially by the National Natural Science Foundation of China (U1505251, 71532014), the National Key Technology R&D Program (2015BAI06B02), the STS Program of the Chinese Academy of Sciences (KFZD-SW-202, KFJ-SW-STS-161), the Guangdong Innovation Research Team Funds for Image-Guided Therapy (2011S013), Key Program for the Guangdong Science and Technology Development Fund (2015B020233011, 2014B010111008), the Shenzhen Science and Technology Development Fund (JCYJ20160229200556939), the Technology Innovation Fund of Shenzhen Peacock Talent (KQJSCX2016030114125686), and Nanshan District Project of Shenzhen (KC2015JSJS0014A). The funders had no role in study design, data collection and analysis, decision to publish, or preparation of the manuscript.

### Grant Disclosures

The following grant information was disclosed by the authors:
National Natural Science Foundation of China: U1505251, 71532014.
National Key Technology R&D Program: 2015BAI06B02.
STS Program of the Chinese Academy of Sciences: KFZD-SW-202, KFJ-SW-STS-161.
Guangdong Innovation Research Team Funds for Image-Guided Therapy: 2011S013.
Key Program for the Guangdong Science and Technology Development Fund: 2015B020233011, 2014B010111008.
Shenzhen Science and Technology Development Fund: JCYJ20160229200556939.
Technology Innovation Fund of Shenzhen Peacock Talent: KQJSCX2016030114125686.
Nanshan District Project of Shenzhen: KC2015JSJS0014A.

### Competing Interests

The authors declare there are no competing interests.

### Author Contributions

- Shengyun Liang analyzed the data, wrote the paper, prepared figures and/or tables.
- Jiali Xu conceived and designed the experiments, performed the experiments, analyzed the data, wrote the paper, prepared figures and/or tables.
- Lei wang contributed reagents/materials/analysis tools, reviewed drafts of the paper.
- Guoru Zhao conceived and designed the experiments, performed the experiments, reviewed drafts of the paper.

### Human Ethics

The following information was supplied relating to ethical approvals (i.e., approving body and any reference numbers):

1. SIAT Institutional Review Board
2. IRB number: SIAT-IRB-160315-H0103.

## Data Availability

The raw data has been supplied as Supplemental Dataset.

## Supplemental Information

Supplemental information for this article can be found online at http://dx.doi.org/10.7717/peerj.2315#supplemental-information.

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
