# Peer review of "An investigation into the bilateral functional differences of the lower limb muscles in standing and walking"

_PeerJ, doi:10.7717/peerj.2315_

## Round 0.1 · original submission · Major Revisions

Both reviewers thinks the topic is interesting. However they raised serious concern for the statistical method applied. The authors should try other choices to make the conclusion solid. In addition, some discussions for the biological meaning are necessary.

Reviewer 1 ·

Basic reporting

No comments

Experimental design

The motivation of the experiment design should be breifly introduced in each step

Validity of the findings

No comments

Additional comments

In this paper, the authors proposed a study about the bilateral functional differences of the lower limb muscles in standing and walking. The following lists my comments.

1. The motivation. The purpose of this study and the rationale of the experimental design should be introduced more carefully. To show the (potential) benefits of the finding differences of low limb muscles, the paper should give more reasons and applications for these differences.
2. The possible reason of the difference. It is good to find out the difference between the muscular activities of the bilateral lower extremities. It is better to list some reasons for the difference.
3. In the discussion, the authors discussed the difference in different people, such as old or young. For robot application, it can be designed according some principles. In this sense, the former comment 2 is important for revealing some tricks of human standing and walking.
4. The statistics. It is strange for me that the p-values are 0. I think it should be <10e-3 or smaller. The authors should carefully check all the p-values.
5. The English should be polished before publication and the expression should be revised carefully for improve the clarity. The figures should be present in higher resolution, such as the legends in Figure 3 (b) and (c).

Reviewer 2 ·

Basic reporting

Thank you for inviting me to review this manuscript. This study recorded sEMG signals from tibialis anterior, soleus, medial gastrocnemius and lateral gastrocnemius muscles of bilateral lower extremities for static steady state and dynamic steady state in order to quantitatively evaluate the sEMG activity asymmetry of bilateral muscles in lower extremities during functional tasks. I think the issue is meaningful and the method by using sEMG is suitable, simple and convenient.
However, I think the manuscript suffered for one serious limit: The statistical method applied in this manuscript is not appropriate. Spearman Correlation analysis is a method that is used to discover the strength of a link between two variables instead of one variable. In this manuscript, the muscle activities for left leg or right leg are the same variable rather than two variables. To analyzing whether themselves have correlation or not is a meaningless issue. Again, comparison and the correlation of different muscle activities are meaningless. I guess logistic regression may be helpful for solving this issue.
Minor suggestion:
1. English: The manuscript needs careful editing English grammar, spelling, and sentence structure so that the goals and results of the study are clear to the reader.
2. The results should be scientifically presented, for example, more raw data should be showed in the tables.

Experimental design

I think the statistical method applied in this manuscript is not appropriate. Spearman Correlation analysis is a method that is used to discover the strength of a link between two variables instead of one variable. In this manuscript, the muscle activities for left leg or right leg are the same variable rather than two variables. To analyzing whether themselves have correlation or not is a meaningless issue. Again, comparison and the correlation of different muscle activities are meaningless. I guess logistic regression may be helpful for solving this issue.

Validity of the findings

Because of the unsuitable statistic method, I don't think the results have a good meaning for solving the issue that the author focus.

Additional comments

Thank you for inviting me to review this manuscript. This study recorded sEMG signals from tibialis anterior, soleus, medial gastrocnemius and lateral gastrocnemius muscles of bilateral lower extremities for static steady state and dynamic steady state in order to quantitatively evaluate the sEMG activity asymmetry of bilateral muscles in lower extremities during functional tasks. I think the issue is meaningful and the method by using sEMG is suitable, simple and convenient.
However, I think the manuscript suffered for one serious limit: The statistical method applied in this manuscript is not appropriate. Spearman Correlation analysis is a method that is used to discover the strength of a link between two variables instead of one variable. In this manuscript, the muscle activities for left leg or right leg are the same variable rather than two variables. To analyzing whether themselves have correlation or not is a meaningless issue. Again, comparison and the correlation of different muscle activities are meaningless. I guess logistic regression may be helpful for solving this issue.
Minor suggestion:
1. English: The manuscript needs careful editing English grammar, spelling, and sentence structure so that the goals and results of the study are clear to the reader.
2. The results should be scientifically presented, for example, more raw data should be showed in the tables.

---

## Round 0.2 · accepted · Accept

The manuscript has been greatly improved after the revision. I suggest its acceptance now.

Reviewer 1 ·

Basic reporting

No Comments

Experimental design

No Comments

Validity of the findings

No Comments

Additional comments

In this version, most of my concerns have been addressed carefully. I have no more critical comments for this paper. One minor suggestion is to revise the figures more clear in higher resolutions (eps or PDF).

Reviewer 2 ·

Basic reporting

The manuscript has been greatly improved. I agree to accept it.

Experimental design

I think now it is more scientific because the authors had changed their statistic methods.

Validity of the findings

Greatly improved.

Additional comments

The manuscript has been greatly improved.